# Establishment and Application of a Quadruplex Real-Time Reverse-Transcription Polymerase Chain Reaction Assay for Differentiation of Porcine Reproductive and Respiratory Syndrome Virus, Porcine Circovirus Type 2, Porcine Circovirus Type 3, and *Streptococcus suis*

**DOI:** 10.3390/microorganisms12030427

**Published:** 2024-02-20

**Authors:** Geng Wang, Hechao Zhu, Cunlin Zhan, Pin Chen, Bin Wu, Zhong Peng, Ping Qian, Guofu Cheng

**Affiliations:** 1College of Veterinary Medicine, Huazhong Agricultural University, Wuhan 430070, China; wanggeng@webmail.hzau.edu.cn (G.W.);; 2Guangxi Yangxiang Co., Ltd., Guigang 537100, China

**Keywords:** multiplex TaqMan PCR, porcine reproductive and respiratory syndrome virus, porcine circovirus type 2, porcine circovirus type 3, *Streptococcus suis*

## Abstract

Respiratory illnesses present a significant threat to porcine health, with co-infections involving Porcine Reproductive and Respiratory Syndrome Virus (PRRSV), *Streptococcus suis* (*SS*), Porcine Circovirus Type 2 (PCV2), and Porcine Circovirus Type 3 (PCV3) acting as the primary causative agents. As a result, the precise diagnosis of PRRSV, PCV2, PCV3 and *SS* is of paramount importance in the prevention and control of respiratory diseases in swine. Therefore, we conducted a molecular bioinformatical analysis to concurrently detect and differentiate PRRSV, PCV2, PCV3 and *SS*. We selected the ORF6 gene of PRRSV, the ORF2 gene of PCV2 and PCV3, and the glutamate dehydrogenase (GDH) gene of *SS* as targets. Specific primers and probes were designed for each pathogen, and following meticulous optimization of reaction conditions, we established a multiple TaqMan fluorescence quantitative PCR detection method. Subsequently, we subjected this method to a comprehensive assessment, evaluating its specificity, sensitivity, and repeatability. The research results demonstrated that the established multiple TaqMan fluorescence quantitative PCR detection method displays displayed exemplary specificity, with no instances of cross-reactivity with other pathogens. The method’s minimum detection concentrations for PRRSV, PCV2, PCV3, and *SS* were 2.80 × 10^1^ copies/µL, 1.96 × 10^2^ copies/µL, 2.30 × 10^2^ copies/µL, and 1.75 × 10^3^ copies/µL, respectively. When applied to the analysis of 30 clinical samples, the results closely mirrored those obtained through Chinese standard uniplex real-time qPCR detection method for PRRSV, as well as the general PCR methods for *SS*, PCV2, and PCV3. This study underscores the robust specificity, high sensitivity, and consistent stability of the multiple TaqMan fluorescence quantitative PCR detection method that we have developed. It is ideally suited to the clinical monitoring of PRRSV, PCV2, PCV3, and *SS*, and it carries significant importance in ongoing efforts to prevent and manage respiratory diseases in porcine populations.

## 1. Introduction

Porcine respiratory disease complex (PRDC) is a significant challenge in the swine industry worldwide [1]. Four common and important respiratory pathogens in swine production include Porcine Reproductive and Respiratory Syndrome Virus (PRRSV), Porcine Circovirus Type 2 (PCV2), Porcine Circovirus Type 3 (PCV3) and *Streptococcus suis* (*SS*) [2,3,4]. Pathogens like PRRSV and PCV2 can cause severe respiratory diseases, suppress the host’s immune function, and create opportunities for secondary pathogens like *Streptococcus* and *Actinobacillus pleuropneumonia* to invade [5,6]. PCV2 and PCV3 often co-infect swine, and their clinical symptoms and pathological changes are similar, making it challenging to differentiate between them. This has led to an increased incidence and severity of swine respiratory diseases, complicating diagnoses [7,8]. Single-pathogen infections may not always result in symptoms, but complex infections involving multiple pathogens can lead to severe diseases [9]. The rapid identification of PRDC’s causative agents is difficult due to multiple pathogen infections. To minimize the economic losses associated with PRDC, itis essential to detect a variety of PRDC pathogens in the herd promptly, accurately, and comprehensively [10,11]. Currently, diagnostic methods for PRRSV, PCV2, PCV3, and *SS* infections include pathogen isolation, conventional or real-time RT-PCR, and other molecular diagnostic techniques [8]. While pathogen isolation is considered the diagnostic confirmation of infection, it has limitations in terms of sensitivity, duration, technical equipment, and expertise, making rapid diagnosis impractical. Additionally, current methods for detecting PRRSV, PCV2, PCV3, and *SS* are mostly based on individual pathogens, lacking the comprehensiveness and speed required to meet the diagnostic needs of Porcine Respiratory Disease Complex (PRDC) [12]. Multiplex TaqMan real-time PCR utilizes specific primers and probes to simultaneously amplify and detect multiple target DNA sequences in a single reaction, leveraging real-time fluorescence monitoring for quantification. This approach offers high sensitivity, strong specificity, and simple operation, providing a rapid and sensitive method for the simultaneous detection of multiple pathogens [13,14].

This study endeavors to establish a TaqMan real-time PCR-based system for the rapid, accurate, and sensitive detection of four respiratory pathogens: PRRSV, PCV2, PCV3, and *SS*, which can simultaneously detect these four viruses in a single reaction without the need for electrophoresis, distinguishing it from traditional PCR methods and showing revolutionary potential for application in pathogen detection and clinical diagnosis. This system will provide rapid, accurate, and reliable technical support for early disease diagnosis and pathogen screening in the region, offering a scientific basis for the development of corresponding prevention and control measures.

## 2. Materials and Methods

### 2.1. Primer and Probe Design

Gene sequences of representative PRRSV2 ORF6, PCV2 and PCV3 ORF2 gene and *SS* GDH deposited in NCBI were analyzed using MEGA7.0. Optimal primers and probes were designed and selected using Primer Premier 5 program. The probe for the ORF6 gene of PRRSV was labeled with the 5′-reported dye 6-carboxyfluorescein (FAM) and the 3′-quencher BHQ1; the probe for the ORF2 gene of PCV2 was labeled with the 5′-reported dye ROX and the 3′-quencher BHQ2; the probe for the ORF2 gene of PCV3 was labeled with the 5′-reported dye VIC and the 3′-quencher BHQ1; and the probe for the GDH gene of *SS* was labeled with the dye CY5 and the 3′-quencher BHQ2. The primers and probes presented in Table 1 were synthesized by Hunan Accurate Biotech Co., Ltd. (Changsha, China).

### 2.2. Standard Plasmid Construction

The ORF6 partial length (125 bp), PCV2 ORF2 (72 bp), PCV3 ORF2 (92 bp), and GDH gene (131 bp) amplified from the positive nucleic acid were cloned into the pMD-18T plasmid (Takara, Beijing, China) to create the recombinant standard plasmids for PRRSV-ORF6-pMD_18_-T (2.80 × 10^10^ copies/µL), PCV2-ORF2- pMD_18_-T (1.96 × 10^10^ copies/µL), PCV3-ORF2-pMD_18_-T (2.30 × 10^10^ copies/µL), and *SS*-GDH- pMD_18_-T (1.75 × 10^10^ copies/µL), respectively. Briefly, RNA was extracted from PRRSV cell cultures and reverse-transcribed into cDNA. DNA from positive samples of *SS*, PCV2, and PCV3 was also extracted. PCR amplification was performed using cDNA from PRRSV and DNA from PCV2, PCV3, and *SS* as templates. Five microliters of the amplification product were analyzed using 1% agarose gel electrophoresis and a gel imaging system. The amplification products were cloned into the pMD-18T vector, and plasmids were extracted and sequenced for validation. These plasmids were used as positive standard plasmids for the experiment and stored at −20 °C after determining their concentration. The copy number of the recombinant plasmids was calculated using the following formula: copy number (copies/μL) = NA (copies/mol) × concentration (g/μL)/MW (g/mol), where NA is Avogadro’s number and MW is the base number times 340.

### 2.3. Optimization of PCR Amplification Conditions

Fluorescent quantitative PCR amplification was performed using plasmids of PRRSV-ORF6-pMD_18_-T, PCV2-ORF2- pMD_18_-T, PCV3-ORF2-pMD_18_-T and *SS*-GDH-pMD_18_-T as templates. The concentrations of primers and probes were optimized as previously described [13]. Briefly, the primer and probe systems were optimized in a 20 μL reaction system, with the addition of 0.2 μL to 1.0 μL of upstream and downstream primers (10 μM) and 0.2 μL to 1.0 μL of the probe (10 μM), TIANGEN^®^ FastKing One-Step RT-qPCR Kit (Probe) (TIANGEN, Beijing, China) 10.8 μL, and nuclease-free water up to 20 μL. The annealing temperature optimization ranged from 50 °C to 60 °C. The quadruplex real-time RT-PCR amplification was performed using the applied LightCycler^®^ 480 Instrument II (Roche, Basel, Switzerland); the amplification condition was set at 94 °C for 30 s, followed by 40 cycles of 94 °C for 5 s, and 60 °C for 30 s; the fluorescent signal was detected at the end of the extension step in each cycle and stored at 25 °C for 300 s.

### 2.4. Construction of Standard Curves

Fluorescent quantitative PCR reactions were performed using 10-fold gradient dilutions (10^−3^–10^−9^) of standard plasmids of PRRSV-ORF6-pMD_18_-T, PCV2-ORF2-pMD_18_-T, PCV3-ORF2-pMD_18_-T and *SS*-GDH-pMD_18_-T as templates under optimized conditions, and the standard curves were then generated. Standard curves were generated based on the cycle threshold (*Ct*) values and the copy numbers (lg values) of the template DNA. Coefficients of determination (*R*^2^) were calculated using GraphPad Prism v. 8.0.1 (https://www.graphpad.com/scientific-software/prism/, accessed on 19 January 2023).

### 2.5. Validation of Specificity, Sensitivity and Stability

The specificity of the generated quadruplex real-time PCR method was validated using the genomic extracted from the other pathogens, including pseudorabies virus (PRV), porcine epidemic diarrhea virus (PEDV), porcine encephalomyelitis virus (PEV), *Pasteurella multocida* (*Pm*), *Staphylococcus aureus* (*S. aureus*), *Escherichia coli* (*E. coli*), and *Glaesserella parasuis* (*G. parasuis*).

Furthermore, to assess the sensitivity of the quadruplex real-time PCR method, *Ct* values were determined by testing different copy numbers (10^−1^–10^4^) of recombinant standard plasmids PRRSV-ORF6-pMD_18_-T, PCV2-ORF2-pMD_18_-T, PCV3-ORF2-pMD_18_-T and *SS*-GDH-pMD_18_-T. We concurrently assessed different virus titers (50% tissue culture infectious dose, TCID_50_) of PRRSV and PCV2, varying bacterial concentrations (CFUs) of *SS* and concentrations of positive PCV3 nucleic acid to determine the minimum quantity of viruses or bacteria detectable with positive signals. Specifically, we extracted nucleic acid from 10^1^–10^6^ TCID_50_ of PRRSV (SinoVet^®^™ Co., Ltd., Tianjin, China, TJ F92 vaccine stains), 10^1^–10^6^ TCID_50_ of PCV2 (PCV2/GX-6 [15]), nucleic acid from 10^1^–10^6^ CFU of *SS*, and 10^1^–10^6^ copies of positive PCV3 nucleic acid, followed by amplification using the quadruplex real-time PCR method. These assays were performed under optimized conditions to test the specificity and sensitivity of the established method.

To test the stability of the generated quadruplex real-time PCR method, separated assays were performed to detect the recombinant standard plasmids at different concentrations and compare the *Ct* values. Briefly, four different concentrations of positive samples were used as templates. Four batches of testing were performed using the multi-fluorescent quantitative PCR method, with each sample set up in triplicate. Intra-group and inter-group repeatability results were used to calculate the coefficient of variation for each.

### 2.6. Comparison with the Uniplex Real-Time PCR Method in Clinical Sample Testing

We also compared the detection results of the quadruplex real-time PCR method developed in this study with uniplex real-time PCR methods in China (according to national or local standards). This was achieved by examining the DNA/RNA extracted from 30 clinical samples; samples including lung tissues were collected from multiple fattening pig farms in several regions (Nanning, Hezhou, Qinzhou, and Yulin) in the Guangxi Zhuang Autonomous Region, China. All the samples were transported at 4 °C and kept at −80 °C for long-term storage. The homogenized lungs and lymph nodes were centrifuged at 6000× *g* for 5 min at 4 °C. Genomic information was extracted using magnetic bead-based DNA/RNA Extraction Kit (NECVB, Harbin, China) following the manufactorer’s instructions. FastKing cDNA First-Strand Synthesis Kits were purchased from Tigen Biotech Co., Ltd. (Beijing, China). A uniplex real-time PCR assay was conducted in a 20.0 μL reaction system, which contained 10.0 μL of TIANGEN^®^ 2 × SuperReal PreMix (Probe) (TIANGEN, Beijing, China); 0.5 μM of the primers for PRRSV, PCV2, PCV3, and *SS*, respectively; and 1.0 μL of template and ddH_2_O to a final volume of 20.0 μL. The amplification program was as follows: pre-denaturation at 95 °C for 15 min, followed by 40 cycles of sequential denaturation at 95 °C for 10 s, annealing at 54 °C for 20 s, and a final extension at 25 °C for 5 min.

## 3. Results

### 3.1. Optimizing Primers and Probes through Sequence Analysis

Sequence comparisons of the primer target regions encompassing PRRSV ORF6, PCV2 and PCV3 ORF2, and *SS* GDH across diverse strains illustrate a high degree of conservation. The analysis depicted in Figure 1 highlights the robust conservation of primers and probes among strains belonging to the same genotype.

### 3.2. Optimization of Quadruplex-qPCR Reaction Conditions

We explored the optimal amplification conditions for the quadruplex real-time PCR method. The total reaction volume for the quadruplex-TaqMan real-time fluorescence quantitative PCR detection method established in this study was 20 µL. Figure 2 shows the impact of different annealing temperatures, probe concentrations, and primer concentrations on fluorescence signals. As depicted in Figure 2, the fluorescence signal of Fam (indicating PRRSV) is strongest at a probe concentration of 0.6 μM (Figure 2A) and a primer concentration of 0.8 μM (Figure 2E). The fluorescence signal of ROX (indicating PCV2) is strongest at a probe concentration of 0.4 μM (Figure 2B) and a primer concentration of 0.6 μM (Figure 2F). The fluorescence signals of VIC (indicating PCV3) and Cy5 (indicating *SS*) are strongest at a probe concentration of 0.6 μM (Figure 2C,D) and a primer concentration of 0.6 μM (Figure 2G,H). When the temperature is 52 °C, the fluorescence intensities of Fam, ROX, VIC, and Cy5 are all maximized (Figure 2I–L). Considering both economic and practical factors, the selected parameters are as follows: the optimal concentrations for the specific primers for PRRSV, PCV3, PCV2, and *SS* are 0.8 μM, 0.6 μM, 0.6 μM, and 0.6 μM, respectively, while the optimal concentrations for the probes are 0.6 μM, 0.4 μM, 0.6 μM, and 0.6 μM, respectively. The annealing temperature of 52 °C yields the best amplification results.

### 3.3. Establishment of Standard Curves

Plasmids with different copy numbers were then detected using quadruplex-PCR. The results revealed the detection limits for PRRSV-ORF6-pMD_18_-T, PCV2-ORF2- pMD_18_-T, PCV3-ORF2-pMD_18_-T and *SS*-GDH-pMD_18_-T (Figure 3A–D). Standard curves plotted using GraphPad Prism software v. 8.0.1 showed that there was a strong linear correlation (*R*^2^ > 0.99) between *Ct* values and the corresponding copy numbers of those plasmids. The standard curves of the four standard plasmids were plotted with slopes of −3.733, −3.558, −3.602 and −4.259, respectively (Figure 3E–G).

### 3.4. Stability, Specificity, and Sensitivity of the Quadruplex Real-Time PCR Method

We chose different positive samples as the cDNA templates to test the coefficient of variation (*C.V.*) values of the method. The results showed that within different detection groups, *C.V.* values were below 1.91%; meanwhile, those between different detection groups were below 3.64% (Table 2), indicating the developed method possesses good stability. Specificity tests revealed that only genomic samples from PRRSV, PCV2, PCV3, and *SS* showed positive amplification curves for the four fluorescence channels of FAM, ROX, VIC, and Cy5; meanwhile, those from PRV, PEDV, PEV, *Pm, S. aureus, E. coli,* and *G. parasuis* did not show amplification curves (Figure 4). In addition, sensitivity tests suggested that detection concentrations for PRRSV, PCV2, PCV3 and *SS* were 2.80 × 10^1^ copies/µL, 1.96 × 10^2^ copies/µL, 2.30 × 10^2^ copies/µL, and 1.75 × 10^3^ copies/µL by testing different copy numbers (10^−1^–10^4^) of those plasmids, respectively (Table 3). By evaluating various virus titers (TCID_50_) of PRRSV and PCV2, different bacterial concentrations (CFU) of *SS*, and varied concentrations of positive PCV3 nucleic acid, this method can detect PRRSV with a minimum limit of 10^2^ TCID_50_, PCV2 with a minimum limit of 10^3^ TCID_50_, and *SS* with a minimum limit of 10^3^ CFU; it also demonstrates high sensitivity to clinically positive PCV3 nucleic acid (Table 4, Figure 5).

### 3.5. Comparison with the Uniplex Real-Time PCR Method in Clinical Sample Testing

To evaluate the accuracy of the quadruplex PCR, genomic nucleic acid was extracted from 30 clinically collected tissue samples suspected of PRRSV, PCV2, PCV3, and *SS* infections. These samples underwent detection using both the quadruplex PCR and conventional gb-PCR. The results demonstrated that out of the 30 clinical samples, the detection rates for PRRSV, PCV2, PCV3, and *SS* using the method developed in this study were 66.67% (20/30), 16.67% (5/30), 73.33% (22/30), and 6.67% (2/30), respectively. Uniplex real-time PCR exhibited detection rates of 66.7% (20/30), 16.67% (5/30), 73.33% (22/30), and 10% (3/30) for PRRSV, PCV2, PCV3, and *SS*, respectively (Table 4). These results suggest that the method developed in this study has a higher detection rate or similar detection results for pathogens compared to uniplex real-time PCR, making it suitable for preliminary clinical testing for PRDC.

## 4. Discussion

PRDC represents a substantial health concern for fattening pigs, encompassing diverse viral and bacterial pathogens [16]. Respiratory ailments in pigs aged 3 to 6 months can arise from infectious and non-infectious factors [11]. Recent advancements in disease control on large-scale swine farms have identified key pathogens contributing to PRDC in pig populations, including PRRSV, PCV2, PCV3, *SS*, and swine pneumonia-like *Mycoplasma* [17]. Co-infections with various PRDC pathogens at varied concentrations are prevalent in clinical settings for fattening pigs [3,4]. The compromised immunity resulting from PRDC pathogen infections in fattening pigs renders them susceptible to additional pathogens, resulting in heightened economic losses [18]. Furthermore, secondary infections from pathogens such as *Actinobacillus pleuropneumoniae, S. suis, G. parasuis,* and *Pasteurella multocida* are associated with PRDC [3,8,19]. Conventional diagnostic methods for PRDC-related viruses involve virus isolation in cell culture, antigen detection through direct fluorescent antibody staining, enzyme immunoassays, and bacterial isolation [3,20]. Despite being cost-effective, these methods are time-consuming, necessitating independent testing for each pathogen [21,22]. Moreover, bacterial pathogen detection relies on culture-based approaches, which may take days to yield results [23]. Due to the high sensitivity and ease of use of PCR and real-time PCR testing, several assays for PRDC-related agents have been developed, but these tests are often specific to individual pathogens [8,24,25]. Although conventional PCR methods are economical, they are less sensitive and require intricate post-PCR processing, limiting their application in sample analysis [21,26]. Different from the conventional PCR, real-time PCR monitors target amplification in real-time, offering heightened sensitivity and rapidity [8]. Recently, Lung et al. reported a novel automated microarray prototype for streamlining post-PCR processing for simultaneous detection and genotyping of bacteria (*Mycoplasma hyopneumonia*, *A. pleuropneumoniae, Salmonella choleraesuis*, and *S. suis*) and viruses (PRRSV, SIV, PCV2, PRCV) [27]. Real-time PCR’s high specificity and sensitivity have proven advantageous in swift pathogen detection [28]. However, a multiplex real-time PCR approach for simultaneous detection of PRRSV, PCV2, PCV3, and *SS* pathogens in pigs is yet to be reported. Our study aims to develop a quadruplex real-time PCR assay capable of simultaneously detecting these pathogens in clinical samples.

Specificity, sensitivity, and stability stand as crucial metrics for diagnostic accuracy [13,26]. In this study, we present a multiplex real-time quantitative PCR detection method adept at identifying multiple PRDC-associated pathogens. This method exhibits high specificity, sensitivity, and operational simplicity. The optimization of multiplex TaqMan real-time PCR necessitates systematic studies and testing under different conditions to identify the optimal combination of annealing temperature, probe concentration, and primer concentration. Critical factors, if deemed significant, were set at the lowest *Ct* value in the final experimental protocol [29]. Primers and probes are pivotal in developing the novel quadruplex real-time PCR method, determining sensitivity and specificity. Proper sequence selection ensures accurate target amplification and reliable fluorescence signal detection, critical for the assay’s success. To optimize them, an initial sequence comparison of the four viruses was conducted, focusing on recent epidemic strains. Primers and probes were designed in each virus’s conservative region (Table 1), and their specificity was confirmed through BLAST analysis on NCBI. In multiplex TaqMan real-time PCR, the concentration of primers and probes significantly impacts reaction efficiency. Optimal primers concentration is crucial for specific target amplification, influencing the overall success and sensitivity. Both excessively high and low probe concentrations can distort signals. Meticulous optimization of probe concentrations is imperative for optimal signal strength. The annealing temperature in multiplex TaqMan real-time PCR critically influences reaction specificity and efficiency, essential for accurate target amplification and reproducible results. Conflicts were resolved based on the number of minimum response parameters and economic applicability. Ultimately, through necessary compromises, we established optimal parameters. Optimal concentrations for specific primers for PRRSV, PCV3, PCV2, and *SS* were set at 0.8 μM, 0.6 μM, 0.6 μM, and 0.6 μM, respectively, while optimal concentrations for probes were 0.6 μM, 0.4 μM, 0.6 μM, and 0.6 μM, respectively. An annealing temperature of 52 °C yielded optimal amplification results (Figure 2).

We chose positive samples as the cDNA/DNA templates to test the coefficient of variation (C.V.) values of the method. The results showed that the within-group C.V. values were below 1.91%, while those between different detection groups were below 3.64%, indicating that the developed method possesses good stability (Table 2). Other viruses and bacteria potentially infecting pigs were employed to assess the quadruplex real-time PCR method’s specificity. The results indicated that primers and probes in the assay neither produced cross-reactions among the four viruses nor elicited nonspecific reactions with other common swine pathogens when all genomic templates existed in the sample pool (Figure 4). Co-infection models gauged the detection efficiency of mixed infections [9]. Sensitivity tests, employing standard curves with synthetic DNA, demonstrated high sensitivity for all primer–probe sets. As shown in Figure 3, the coefficient of determination (*R*^2^) exceeded 0.99 in the range of 10^7^ copies/μL to 10^1^ copies/μL, indicating the quantitative range of this method. The Ct limit value, defining the lowest copy number yielding a detectable PCR amplification product at least 95% of the time, was set at 36, 34, 36, and 33 for PRRSV, PCV2, PCV3, and *SS*, respectively. This corresponds to a detection limit of 2.80 × 10^1^ copies/µL, 1.96 × 10^2^ copies/µL, 2.30 × 10^2^ copies/µL, and 1.75 × 10^3^ copies/µL as the limit concentration (Table 3). Given that amplification of the target gene on recombinant plasmids is typically easier than amplification from the genome or cDNA, we assessed the method by examining various virus titers (TCID_50_) of PRRSV and PCV2, different bacterial concentrations (CFU) of *SS*, and varied concentrations of positive PCV3 nucleic acid. Our findings reveal that the method is capable of detecting PRRSV at a minimum limit of 10^2^ TCID_50_, PCV2 at a minimum limit of 10^2^ TCID_50_, and *SS* at a minimum limit of 10^2^ CFU, demonstrating heightened sensitivity towards clinical positive PCV3 nucleic acid (Table 4, Figure 5). The findings suggest that the quadruplex real-time PCR method is accurate and applicable for analyzing clinical samples from fattening pigs affected by PRDC.

We compared the detection results of the quadruplex real-time PCR method developed in this study with uniplex real-time PCR methods in China (according to national or local standards). These findings indicate that the method developed in this study either exhibits a higher detection rate or yields similar detection results for pathogens when compared to uniplex real-time PCR, rendering it suitable for preliminary clinical testing of PRDC (Table 5). The developed method has been applied for a pilot study of clinical samples (lung, inguinal lymph nodes, mesenteric lymph nodes, hilar lymph nodes), and all these pathogens were detected. It is worth noting that the analysis of clinical samples indicates a high positive infection rate of PRRSV and PCV3 in the Guangxi Zhuang Autonomous Region fattening pig market compared to PCV2 and *SS*. This situation emphasizes the importance of monitoring PRDC pathogens. In this regard, the method developed in the study holds significant practical value.

The quadruplex real-time quantitative PCR test established in this study comprehensively and simultaneously detects multiple PRDC pathogens, rapidly confirming their presence in samples. Moreover, the quadruplex PCR method, detecting four common pathogens in a single reaction, substantially reduces costs compared to individual PCRs or uniplex real-time PCR, thus enhancing overall work efficiency [30]. This has important implications for the prevention and control of these four diseases in clinical practice.

## 5. Conclusions

The developed multiplex RT-PCR method exhibits excellent specificity, efficient detection capabilities, and applicability to laboratory diagnosis, epidemiological research, and monitoring of PRRSV, PVC2, PCV3, and *SS*. Moreover, the method is suitable for clinical differential diagnosis of mixed infections and facilitates early diagnosis of clinical cases.

## Figures and Tables

**Figure 1 microorganisms-12-00427-f001:**
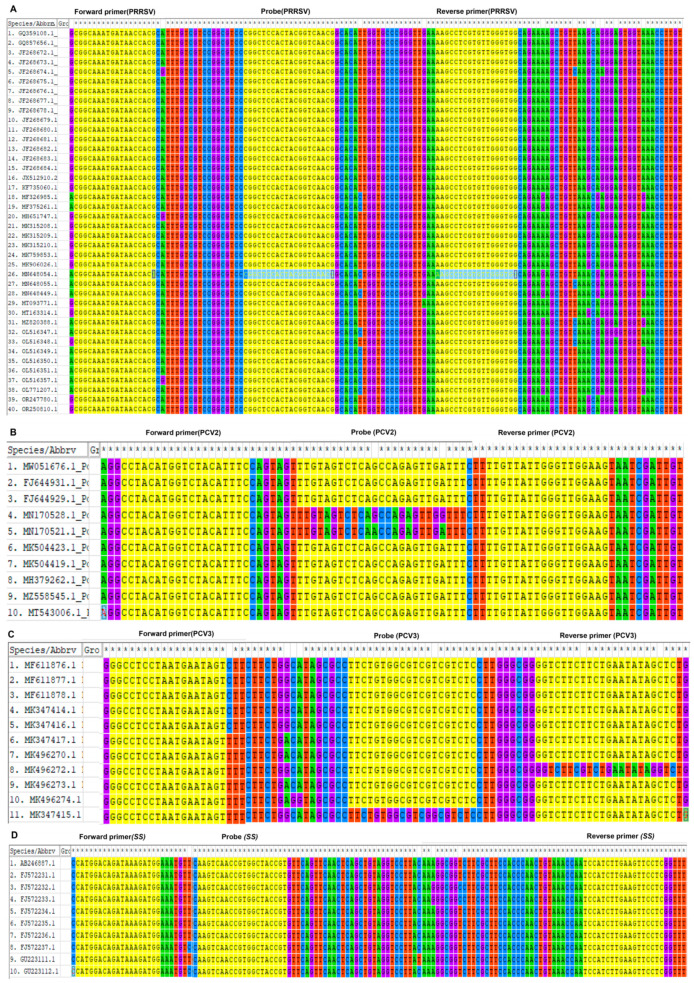
**Nucleotide sequence comparisons of the primers and probes target regions of the four pathogens.** TaqMan primer and probe regions showing matches in PRRSV (**A**), PCV2 (**B**), PCV3 (**C**), and *SS* (**D**). The yellow highlighted areas are the locations of primers and probes.

**Figure 2 microorganisms-12-00427-f002:**
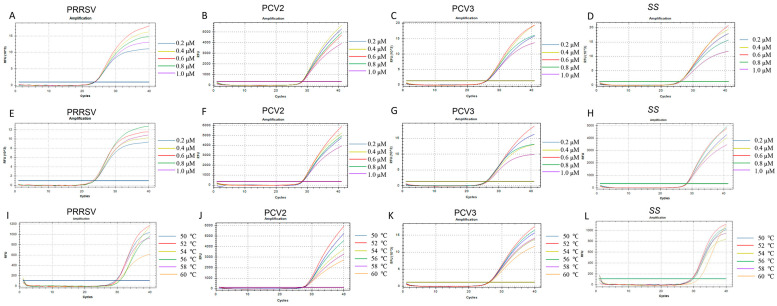
**Optimization of fluorescence intensity.** The effect of different probe concentrations on the fluorescence signal of PRRSV (**A**), PCV2 (**B**), PCV3 (**C**), and *SS* (**D**). The effect of different primer concentrations on the fluorescence signal of PRRSV (**E**), PCV2 (**F**), PCV3 (**G**) A, and *SS* (**H**). The effect of different annealing temperatures on the fluorescence signal of PRRSV (**I**), PCV2 (**J**), PCV3 (**K**) A, and *SS* (**L**).

**Figure 3 microorganisms-12-00427-f003:**
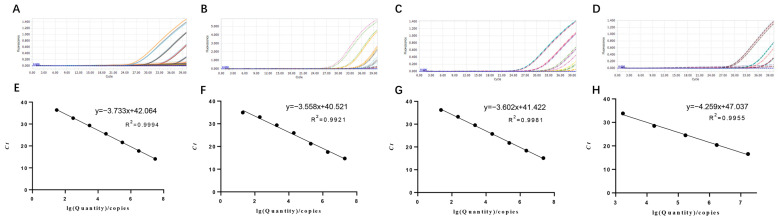
**Detection limit and standard curves of the quadruplex real-time PCR method developed in this study.** (**A**) Detection limit for OFR6 gene of PRRSV; (**B**) detection limit for ORF2 gene of PCV2; (**C**) detection limit for ORF2 gene of PCV3; (**D**) detection limit for GDH gene of *SS.* (**E**) Plasmid DNA standard curve for OFR6 gene of PRRSV, y = −3.733x + 42.064, *R*^2^ = 0.999, E = 85.39%; (**F**) plasmid DNA standard curve for ORF2 gene of PCV2, y = −3.558x + 40.521, *R*^2^ = 0.999, E = 90.98%; (**G**) plasmid DNA standard curve for ORF2 gene of PCV3, y = −3.602x + 41.422 *R*^2^ = 0.998, E = 89.49%; (**H**) plasmid DNA standard curve for GDH gene of *SS*, y = −4.259x + 47.037, *R*^2^ = 0.996, E = 71.70%.

**Figure 4 microorganisms-12-00427-f004:**
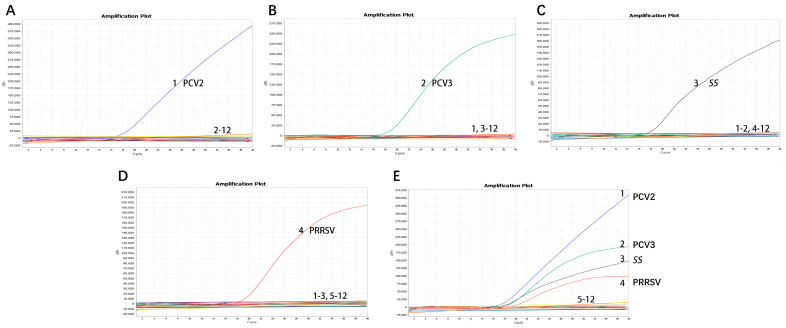
**Specificity of the quadruplex real-time PCR assay.** (**A**) Amplification curve of PCV2 (only PCV2-positive nucleic acids); (**B**) amplification curve of PCV3 (only PCV3-positive nucleic acids); (**C**) amplification curve of *SS* (only *SS*-positive nucleic acids); (**D**) amplification curve of PRRSV (only PRRSV-positive nucleic acids); (**E**) amplification curve of PRRSV, PCV2, PCV3 and *SS*. 1: PCV2-positive nucleic acid; 2: PCV3-positive nucleic acid; 3: *SS*-positive nucleic acid; 4: PRRSV-positive nucleic acid; 5: PEDV-positive nucleic acid; 6, PRV-positive nucleic acid; 7: PEV-positive nucleic acid; 8: *G. parasuis*-positive nucleic acid; 9: *Pm*-positive nucleic acid; 10: *S. aureus*-positive nucleic acid; 11: *E.coli*-positive nucleic acid; 12: ddH_2_O.

**Figure 5 microorganisms-12-00427-f005:**
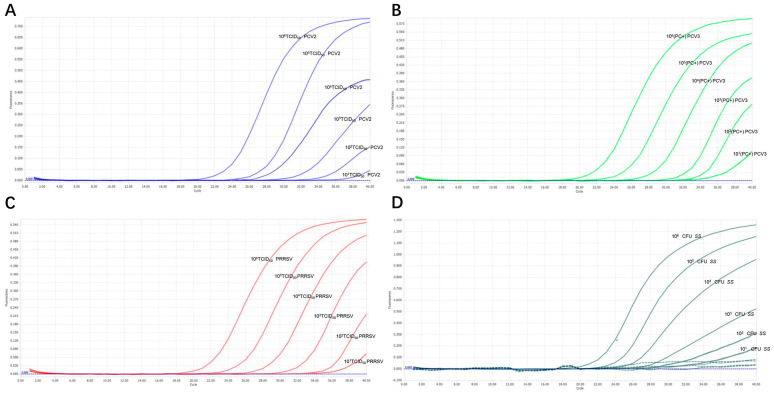
**Detection limit of the four pathogens using the quadruplex real-time PCR method.** (**A**) Amplification curve of PCV2 by different virus titers (TCID_50_); (**B**) amplification curve of different concentrations of positive PCV3 nucleic acid (PC+); (**C**) amplification curve of different bacterial concentrations (CFU) of *SS*; (**D**) amplification curve of PRRSV with different virus titers (TCID_50_).

**Table 1 microorganisms-12-00427-t001:** Primers and probes used in this study. F, R, and P indicate forward primer, reverse primer, and probe, respectively.

Name	Gene	Sequence (5′-3′) of Primer/Probe	Size (bp)
PRRSV-F	ORF6	CGGCAAATGATAACCACG	
PRRSV-R		CCACCCAACACGAGGCTT	125 bp
PRRSV-P		FAM-CGGCTCCACTACGGTCAACG-BHQ1	
PCV2-F	ORF2	CCTACATGGTCTACATTTC	
PCV2-R		CTTCCAACCCAATAACAA	72 bp
PCV2-P		ROX-AATCAACTCTGGCTGAGACTACAA-BHQ2	
PCV3-F	ORF2	GGCCTCCTAATGAATAGTC	
PCV3-R		GAGCTATATTCAGAAGAAGACC	92 bp
PCV3-P		VIC-TTCTGTGGCGTCGTCGTCTC-BHQ1	
*SS*-F	GDH	CATGGACAGATAAAGATGG	
*SS*-R		GAGGAACTTCAAGATGGA	131 bp
*SS*-P		CY5-AAGTCAACCGTGGCTACCGT-BHQ2	

**Table 2 microorganisms-12-00427-t002:** Validation of the detection repeatability of the developed quadruplex real-time PCR method.

Pathogens	DNA/cDNA (Positive Samples)	Within-Group Test		Between-Group Test	
		*Ct* ^a^ (Mean ± SD)	*C.V.* ^b^	*Ct* (Mean ± SD)	*C.V.* ^b^
PRRSV	1	21.60 ± 0.18	0.84%	21.80 ± 0.21	0.92%
	2	17.71 ± 0.15	0.83%	17.95 ± 0.24	1.36%
	3	13.79 ± 0.06	0.47%	13.94 ± 0.15	1.08%
PCV2	1	21.22 ± 0.080	0.39%	21.92 ± 0.69	3.18%
	2	17.56 ± 0.01	0.07%	18.07 ± 0.50	2.81%
	3	13.76 ± 0.09	0.06%	14.93 ± 0.47	3.31%
PCV3	1	21.75 ± 0.19	0.89%	22.00 ± 0.24	1.12%
	2	18.39 ± 0.35	1.91%	19.00 ± 0.61	3.26%
	3	14.05 ± 0.22	1.57%	14.58 ± 0.53	3.64%
*SS*	1	24.51 ± 0.21	0.86%	24.56 ± 0.14	0.57%
	2	20.39 ± 0.22	1.60%	20.45 ± 0.15	0.73%
	3	16.56 ± 0.06	0.39%	16.36 ± 0.20	1.24%

^a^ *Ct*, cycle threshold; ^b^ *C.V.*, coefficient of variation.

**Table 3 microorganisms-12-00427-t003:** The detection limits of quadruplex real-time PCR methods detecting the different concentrations of those standard plasmids.

Plasmid Name	10-Fold Gradient Dilutions	10^4^	10^3^	10^2^	10^1^	10^0^	10^−1^
PRRSV-ORF6-pMD18T	concentrations (copies/μL)	2.80 × 10^4^	2.80 × 10^3^	2.80 × 10^2^	2.80 × 10^1^	2.8 × 10^0^	2.80 × 10^−1^
	*Ct* value	25.55	29.35	32.7	36.37	N/A ^a^	N/A
PCV2-ORF2-pMD18T	concentrations (copies/μL)	1.96 × 10^4^	1.96 × 10^3^	1.96 × 10^2^	1.96 × 10^1^	1.96 × 10^0^	1.96 × 10^−1^
	*Ct* value	26.66	29.39	32.94	34.86	N/A	N/A
PCV3-ORF2-pMD18T	concentrations (copies/μL)	2.30 × 10^4^	2.30 × 10^3^	2.30 × 10^2^	2.30 × 10^1^	2.30 × 10^0^	2.30 × 10^−1^
	*Ct* value	25.7	29.56	33.26	36.22	N/A	N/A
*SS*-GDH-pMD18T	Concentrations (copies/μL)	1.75 × 10^4^	1.75 × 10^3^	1.75 × 10^2^	1.75 × 10^1^	1.75 × 10^0^	1.75 × 10^−1^
	*Ct* value	28.51	33.81	N/A	N/A	N/A	N/A

^a^ N/A indicated “No *Ct*” value.

**Table 4 microorganisms-12-00427-t004:** The detection limits of quadruplex real-time PCR methods detecting the different concentrations of the four pathogens.

Positive Nucleic Acid	10-Fold Gradient Dilutions	10^6^	10^5^	10^4^	10^3^	10^2^	10^1^
PRRSV (vaccine of TJ-F92 strains)	Virus tites(TCID_50_/mL)	10^6^ TCID_50_	10^5^ TCID_50_	10^4^ TCID_50_	10^3^ TCID_50_	10^2^ TCID_50_	10^1^ TCID_50_
	*Ct* value	19.52	23.34	27.10	30.94	35.25	N/A ^a^
PCV2 (PCV2-GX-6 strains)	Bacterial count by plate (TCID_50_/mL)	10^6^ TCID_50_	10^5^ TCID_50_	10^4^ TCID_50_	10^3^ TCID_50_	10^2^ TCID_50_	10^1^ TCID_50_
	*Ct* value	24.20	27.60	30.14	34.49	N/A	N/A
PCV3 (Positive nucleic acid)	10-fold gradient dilutions	10^6^	10^5^	10^4^	10^3^	10^2^	10^1^
	*Ct* value	19.72	23.03	27.07	30.98	34.05	N/A
*SS*	Bacterial count by plate (CFU/mL)	10^6^ CFU	10^5^ CFU	10^4^ CFU	10^3^ CFU	10^2^ CFU	10^1^ CFU
	*Ct* value	20.12	24.01	27.56	31.79	N/A	N/A

^a^ N/A indicated “No *Ct*” value.

**Table 5 microorganisms-12-00427-t005:** Results of quadruplex real-time PCR methods detecting the clinical samples.

Samples	Types	PRRSV		*SS*		PCV2		PCV3	
		qu-PCR ^c^	gb-PCR ^a^	qu-PCR ^c^	gb-PCR ^a^	qu-PCR ^c^	gb-PCR ^a^	qu-PCR ^c^	db-PCR ^b^
1	lung	+	+	+	+	+	+	+	+
2	lung	+	+	−	−	−	−	+	+
3	lung	+	+	−	−	−	−	+	+
4	lung	−	−	−	+	−	−	−	−
5	lung	+	+	−	−	−	−	+	+
6	serum	−	−	−	−	−	−	+	+
7	serum	+	+	−	−	−	−	−	−
8	serum	+	+	−	−	−	−	+	+
9	serum	−	−	−	−	−	−	+	+
10	serum	+	+	−	−	−	−	−	−
11	spleen	−	−	−	−	−	−	−	−
12	spleen	+	+	−	−	−	−	+	+
13	spleen	+	+	−	+	+	+	+	+
14	spleen	+	+	−	−	−	−	+	+
15	tonsil	−	−	−	−	−	−	+	+
16	tonsil	+	+	−	−	−	−	+	+
17	tonsil	−	−	−	−	−	−	+	+
18	tonsil	+	+	+	+	−	−	+	+
19	MLN	+	+	−	−	−	−	−	−
20	MLN	+	+	−	−	−	−	+	+
21	MLN	+	+	−	−	−	−	+	+
22	MLN	−	−	−	−	+	+	+	+
23	ILN	+	+	−	−	−	−	−	−
24	ILN	+	+	−	−	−	−	+	+
25	ILN	−	-	−	−	−	−	+	+
26	ILN	+	+	−	−	+	+	+	+
27	HLN	+	+	−	−	−	−	−	−
28	HLN	−	−	−	−	−	−	+	+
29	HLN	+	+	−	−	−	−	+	+
30	HLN	−	−	−	−	+	+	−	−

^a^ gb-PCR: Chinese standard uniplex real-time PCR for the diagnosis of PRRSV (GBT18090-2023), ss(GBT19915.6-2005) and PCV2(GBT35901-2018), respectively. ^b^ db-PCR: Local standard uniplex real-time PCR for the diagnosis of PCV3 (DB31T955-2022). ^c^ qu-PCR: Quadruplex real-time PCR method developed in this study. “+” refers to positive; “−” refers to negative; ILN indicates inguinal lymph nodes; MLN indicates mesenteric lymph nodes; and HLN indicates hilar lymph nodes.

## Data Availability

All data supporting the results of this study are included in the manuscript.

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
