# Peer review of "Establishment and Application of a Quadruplex Real-Time Reverse-Transcription Polymerase Chain Reaction Assay for Differentiation of Porcine Reproductive and Respiratory Syndrome Virus, Porcine Circovirus Type 2, Porcine Circovirus Type 3, and Streptococcus suis"

_microorganisms, 2024, doi:10.3390/microorganisms12030427_

Round 1
Reviewer 1 Report
Comments and Suggestions for Authors
The presented work has a novelty however needs better representation. Following points need to be considered:
-The introduction section should be in further detail about the drawbacks of existing methods and how the proposed multiplex PCR method provides advantages over current methods.
-Introduction section does not have a explanation about following abbreviation: TaqMan real-time PCR,
-The discussion regarding the impact of different annealing temperatures, probe concentrations, and primer concentrations on fluorescence signals is not enough. More discussion should be added for the optimization experiments.
-Figure 3: Serious resolution problem. Please upload figures with better resolution. B and F is missing in the figure. The experimental point in E,F,G and H represent one repetition? It is ideal to have at least three repetition and error bars in the calibration figures.
-Not enough discussion for figure 4. Please make it clear on the figure that there is a visible difference between the interferences and targets. Refer the figure 4 quantitatively to support the discussion.
-Bed resolution of Figure 4. Please make a clear indication of what the numbers 1 2 3 4 represent on the figure.
-There is no discussion about Table 4.
Comments on the Quality of English LanguageEditting is required.
Author Response
Reviewer #1 (Comments for the Author):
The presented work has a novelty however needs better representation. Following points need to be considered:
-The introduction section should be in further detail about the drawbacks of existing methods and how the proposed multiplex PCR method provides advantages over current methods.
Response: Thank you for your valuable suggestion. Lines 58-71, we have elucidated the drawbacks of existing methods and outlined the advantages of the proposed multiplex PCR method in the revision.
-Introduction section does not have a explanation about following abbreviation: TaqMan real-time PCR,
Response: Thank you for your valuable suggestion. Lines 58-63, the sentence has been revised to “Multiplex TaqMan real-time PCR utilizes specific primers and probes to simultaneously amplify and detect multiple target DNA sequences in a single reaction, leveraging real-time fluorescence monitoring for quantification. This approach offers high sensitivity, strong specificity, and simple operation, providing a rapid and sensitive method for the simultaneous detection of multiple pathogens” in the revision.
-The discussion regarding the impact of different annealing temperatures, probe concentrations, and primer concentrations on fluorescence signals is not enough. More discussion should be added for the optimization experiments.
Response: Thank you for your valuable suggestion. Lines 290-308, we have added more discussion on the impact of primer-probe sequences, different annealing temperatures, and the concentrations of probes and primers on fluorescence signals in the revision.
-Figure 3: Serious resolution problem. Please upload figures with better resolution. B and F is missing in the figure. The experimental point in E,F,G and H represent one repetition? It is ideal to have at least three repetition and error bars in the calibration figures.
Response: Thank you for your valuable suggestion. We have re-uploaded the high-resolution Figure 3. Standard curves in E, F, G, H were repeated three times, but with the coefficient of determination (R2) values close to 1, the three repeated points overlap, and the error bars are not clearly evident.
-Not enough discussion for figure 4. Please make it clear on the figure that there is a visible difference between the interferences and targets. Refer the figure 4 quantitatively to support the discussion. Bed resolution of Figure 4. Please make a clear indication of what the numbers 1 2 3 4 represent on the figure.
Response: Thank you for your valuable suggestion. The amplification curves in fluorescence quantitative PCR vividly demonstrate the distinction between target pathogens and interfering pathogens (non-target). Only the amplification of target pathogens produces a specific S-shaped curve, while other interferences (non-target pathogens) result in a different pattern, providing a clear visual representation of the results for readers. Additionally, the numbers in Figure 4 correspond to the pathogens, as explained in the legend: 1, PCV2 (positive nucleic acid); 2, PCV3 (positive nucleic acid); 3, SS (positive nucleic acid); 4, PRRSV (positive nucleic acid); 5, PEDV (positive nucleic acid); 6, PRV (positive nucleic acid); 7, PEV (positive nucleic acid); 8, G. parasuis (positive nucleic acid); 9, Pm (positive nucleic acid); 10, S. aureus (positive nucleic acid); 11, E. coli (positive nucleic acid); 12, ddH2O.
-There is no discussion about Table 4.
Response: Thank you for your valuable suggestion. Lins 336-348, the discussion about Table 4 has been revised in the revision.

Reviewer 2 Report
Comments and Suggestions for Authors
The manuscript presents a molecular study for the simultaneous detection of the Porcine Reproductive and Respiratory Syndrome Virus (PRRSV), Streptococcus suis (SS), Porcine Circovirus Type 2 (PCV2), and Porcine Circovirus Type 3 (PCV3) using RT-PCR. The Authors developed a multiple TaqMan fluorescence quantitative PCR detection method selecting the ORF6 gene of PRRSV, the ORF2 gene of PCV2 and PCV3 and the glutamate dehydrogenase (GDH) gene of SS as targets. This study is practically a technical paper, for the development of a molecular assay. The Authors find the developed quadruple detection method very specific, sensitive, and stable for clinical applications. However, there are some details below that are lacking from this manuscript and should be added.
1. The Authors compared the detection results of the quadruplex real-time PCR method that they developed in this study with conventional PCR methods (3.4. Comparison with Conventional PCR method in Clinical Sample Testing). The results were found comparable, with the quadruplex method producing higher or similar detection results to the conventional PCR results (Table 3). Assuming that these amplifications were carried out by the Authors, it would be helpful to include the results of the gel electrophoresis separation of these amplicons in the supplementary information.
2. The qPCR results of this study are validated by nucleic acid-based conventional PCR results which are very similar assays. The real validation should use plating, cell culture or microscopy-based counts to compare the number of target molecules present to the number calculated based on the qPCR results.
3. What is the minimum number of target molecules/µL that can be detected by the quadruplex qPCR method?
Author Response
Reviewer #2 (Comments for the Author):
The manuscript presents a molecular study for the simultaneous detection of the Porcine Reproductive and Respiratory Syndrome Virus (PRRSV), Streptococcus suis (SS), Porcine Circovirus Type 2 (PCV2), and Porcine Circovirus Type 3 (PCV3) using RT-PCR. The Authors developed a multiple TaqMan fluorescence quantitative PCR detection method selecting the ORF6 gene of PRRSV, the ORF2 gene of PCV2 and PCV3 and the glutamate dehydrogenase (GDH) gene of SS as targets. This study is practically a technical paper, for the development of a molecular assay. The Authors find the developed quadruple detection method very specific, sensitive, and stable for clinical applications. However, there are some details below that are lacking from this manuscript and should be added.
1. The Authors compared the detection results of the quadruplex real-time PCR method that they developed in this study with conventional PCR methods. The results were found comparable, with the quadruplex method producing higher or similar detection results to the conventional PCR results (Table 3). Assuming that these amplifications were carried out by the Authors, it would be helpful to include the results of the gel electrophoresis separation of these amplicons in the supplementary information.
Response: Thank you for your valuable suggestion. We sincerely apologize for any writing errors. In our study, we compared with uniplex real-time quantitative PCR rather than validating the results using gel electrophoresis.
2. The qPCR results of this study are validated by nucleic acid-based conventional PCR results which are very similar assays. The real validation should use plating, cell culture or microscopy-based counts to compare the number of target molecules present to the number calculated based on the qPCR results.
Response: Thank you for your valuable suggestion. However, given the intricate nature of PRRSV strains, quantifying TCID50 for PCV2 (which lacks CPE) and isolating the virus for PCV3 present challenges, hindering direct comparisons of the aforementioned methods. Consequently, this study opted to employ standard plasmids with known concentrations to establish the minimum detectable amount.
3. What is the minimum number of target molecules/µL that can be detected by the quadruplex qPCR method?
Response: Thank you for your valuable suggestion. Many studies have indicated that employing standard plasmids for nucleic acid quantification establishes a minimum detection limit, making it more comprehensible and readily applicable for readers. In addition, sensitivity tests suggested that detection concentrations for PRRSV, PCV2, PCV3 and SS are 2.80×101 copies/µL, 1.96×102 copies/µL, 2.30×102 copies/µL, and 1.75×103 copies/µL by testing different copy numbers (10-1-104) of those plasmids, re-spectively(Table 3)

Round 2
Reviewer 1 Report
Comments and Suggestions for Authors
Authors improved the manuscript. I recommend for publication.
Author Response
The reviewer indicated that there are no suggested modifications.
Reviewer 2 Report
Comments and Suggestions for Authors
Thank you for your responses. Please provide the minimum number of viruses or bacteria that would give a detectable positive signal similar to those in Figure 4 using this quadruplex assay.
It is easy to amplify a target gene on a recombinant plasmid (Table 3) compared to a genomic or cDNA template.
Author Response
Please provide the minimum number of viruses or bacteria that would give a detectable positive signal similar to those in Figure 4 using this quadruplex assay.
It is easy to amplify a target gene on a recombinant plasmid (Table 3) compared to a genomic or cDNA template.
Response: Thank you for your valuable suggestion. Line 247-254, we have utilized this method to assess the detection sensitivity of various virus titers (TCID50) of PRRSV and PCV2, different bacterial concentrations (CFU) of SS, and varied concentrations of positive PCV3 nucleic acid. this method can detect PRRSV with a minimum limit of 102 TCID50, PCV2 with a minimum limit of 103 TCID50, SS with a minimum limit of 103 CFU, and demonstrates high sensitivity to clinical positive PCV3 nucleic acid in the revision(Table 4, Figure 5, seen below panels) .
